# Time-Space Compression Effect of High-Speed Rail on Tourist Destinations in China

**Taohong Li** [1,*]**, Hong Shi** [2]**, Ning Chris Chen** [3] **and Luo Yang** [2]

1   Business and Tourism School, Sichuan Agricultural University, Dujiangyan 611800, China
2   School of Tourism and Historical Culture, Southwest Minzu University, Chengdu 610093, China
3   Department of Management, Marketing, and Entrepreneurship, University of Canterbury, Christchurch 8140, New Zealand
*   Correspondence: 41506@sicau.edu.cn; Tel.: +86-187-8227-6173

**Abstract:** This study proposes a time-space compression (TSC) model and evaluates the TSC effect of high-speed rail (HSR) on a sample of 2662 classified tourist destinations from 2008 to 2019 in China with the help of GIS technology. Based on panel models, this study finds that, within five hours: (1) the TSC effect of HSR on tourist destinations in eastern and central China is three times stronger than that in western and north-eastern China; (2) the negative impact coefficient of TSC of HSR on tourist destination development in China within temporal distances (3 h, 4 h, 5 h) are $-0.193$, $-0.117$, and $-0.091$ respectively; and (3) the farther the temporal distance, the weaker the inhibitory effect. Results from this study contribute to the literature by providing empirical evidence of the potentially negative TSC effect on regional and tourism development. Findings provide managerial implications suggesting that tourist destinations should implement marketing policies to retain tourists and prevent the loss of tourists brought by the opening of HSR.

**Keywords:** high-speed rail; tourist destinations; time-space compression effect; GIS; inhibitory effect; China

## 1. Introduction

In early 19th century, Harvey [1] used the term "annihilation of time by space" to describe the new reality of mobility, based on revolutionary technological development in transportation. Physical distance stopped being a prerequisite factor in the consideration of mobility, but a subjective feeling about time–space [2], namely the individual's experience of time and space under the influence of sensation and perception. In recent years, the endowment of quality transport infrastructure, such as HSR, is regarded as a prior condition for the experience, as well as the material basis of travelers' perception of space and time.

Modern transportation facilities and accessibility are important prerequisites for the healthy development of tourist destinations. Khadaroo and Seetanah [3] recognized that, on top of tourism infrastructure and other classical determinants, transport infrastructure is a significant determinant for tourism inflows into a destination. From a technical point of view, the HSR has been proven to be a safe, comfortable, and efficient mode of transportation [4]. As a more competitive and attractive mode of transport [5,6] and a sustainable initiative [7], HSR became a vital service for hub cities and promoted the development of business and leisure tourism in cities [8,9]. For example, the Spanish HSR, known as AVE (Alta Velocidad Española), is widely regarded as a success [10], and Japan's Shinkansen system, also referred to as "HSR corridors", greatly reduces travel time [11]. The TGV (Train à Grande Vitesse) services benefit the well-known tourist destinations in France, and the availability of TGV adds value to tourism development [12]. China's HSR is also widely regarded as a success, since it has emerged one of residents' most important travel modes and improved the service standards and the technical quality of infrastructure [13].

Studies on the impact of HSR on the spatial structure of tourism can be traced back to the 1990s. Results from Gutiérrez et al. [14] showed that the opening of HSR has improved the overall tourism accessibility in Europe. In the literature, accessibility has been the main research perspective to study the relationship between HSR and the spatial structure of regional tourism, including potential accessibility [15] and daily accessibility [16]. There is a general consensus that HSR reduces travel costs, shortens travel distance, changes urban accessibility, and thus changes the tourist destinations. Some existing studies investigate effects of HSR in the scope of TSC. However, most existing studies focused solely on tourist destination regions, without examining the physical distance of certain famous scenic spots. Given that famous scenic spots representing tourist destinations should be regarded as the starting point with respect to the TSC effect, to study TSC effect caused by HSR, the actual physical location of famous scenic spots should not be ignored. Tourist markets within a relatively short distance may demonstrate entirely different reactions toward applying HSR to access tourist destinations. In addition, the introduction of HSR not only brings certain tourist generating regions closer to a tourist destination, but also reaches new tourist markets which were not previously accessible via public transportation. From a research method point of view, tourism scholars adopted some policy methods and econometric models to study the impact of the HSR, such as the DID and the PSM–DID [17,18]. However, limited studies have specifically estimated the extent of the TSC effect of tourist destinations under the impact of HSR.

To fill in these gaps, this study reviews the literature regarding HSR development and its impact on tourism, as well as theoretical developments around TSC, and proposes a TSC model for tourist destination development in terms of temporal distance. The TSC effect is further conceptualized in two forms: time compression (TC) effect and spatial expansion (SE) effect. Through a dataset on a sample of 2662 AAAA and AAAAA classified tourist destinations in China, this study assesses the TSC effect of HSR on tourist destinations.

## 2. Literature Review

### 2.1. Time-Space Compression

Time and space are so "natural" that they are the basic categories of human existence [19,20]. Within the discipline of geography, time-space compression has been widely addressed in the literature, specifically in the last decade. Janelle [21] proposed the concept of "time-space convergence" to describe the rate at which travelling time is reduced due to transport and communication innovation. TSC, as advanced by Harvey [22], is rather distinct from time-space convergence; it not only has the connotation of time-space convergence, but also underscores the ability of technology (railways, expressways, airlines, etc.) to reduce spatial barriers, to annihilate space by time. As a Marxist, Harvey explores the intersecting command of money, time, and space, which form a fundamental source of social power. He found that the revolutionary qualities of a capitalistic mode of production, marked by strong currents of technological change and rapid economic growth and development, have been associated with powerful revolutions in the social conceptions of space and time [23]. Hereby, as a result of the "annihilation of space by time" and the compression of time and space, on the one hand, the pace of people's lives is accelerated, and the space barriers are removed. On the other hand, the communication in politics, economy, and culture of various regions in the world is accelerated, thus resulting in the current globalization. The fields of culture, politics, and society are finally brought home within the experience of TSC as a new research perspective and a stimulus for reconsidering the reconstruction of industries in contemporary social life.

Some authors discussed how society has compressed time and space even they are slippery topics [20]. Additional aspects, such as complete time-space convergence [24], the significant time-space convergence of accessibility [25], intercity connectivity [26], or mobility [27], have also been added into discussion. Particularly, speed, especially the struggle for time, is intricately linked with TSC [28], based on which a significant amount of literature concerning temporal and spatial dimensions has been developed. In the

ongoing effort to reduce the resistance of space, time turns into a form of currency once it is instrumentalized [29]. Other studies have focused on issues, such as transportation and technology. Progress, such as transport and communications technologies, contribute to the conquest of space [30], the overcoming of all spatial barriers [31], and the ultimate "annihilation of space through time" [19]. People with free access to the connecting grid can produce more products and reach farther places within a given time. The effect is that, on the one hand, the time and space experienced by people are "compressed", on the other hand, people have more time and a larger scope of activities [23,32].

There are several contributions in the literature concerning transport modes and TSC, such as rail, HSR, and air transportation [33], which make the world seem smaller [34]. Whether it is transportation technology, capital accumulation, physical replication, or spatial folding, these factors are important to understanding the compression of space and time [35]. Each act of TSC undermines the earlier geographical location and creates a new geographical location. Moreover, the process is entangled with class, gender, and race relations across all societies. In this sense, TSC is concerned with the construction and reconstruction of relational geographies into an interconnected system that bind people and places unevenly in a changing landscape, in which time and space rotate and fold and rotate in complex, accidental ways.

To be clear, there is no single theory that can explain TSC in different historical and spatial contexts. Because TSC takes different forms, shows different patterns, reflects different causes, and implies different outcomes, it thus depends on where and when it happens and who is involved [20]. From the perspective of relational space, the TSC includes TC and SE. In the process of tourism development, the TSC effect will be produced due to the improvement of transportation facilities. It is necessary to explore the TSC effect of tourist destinations from the perspective of tourism system.

### 2.2. TSC Effect of HSR on Tourist Destinations

Tourism geography scholars have argued that tourism should be positioned within the general context of human mobility, considering a higher level of mobility instead of focusing on a purely consumer decision approach to understanding travelling behaviors [36]. Even studying tourist experience in tourist destinations relies on tourists' mental and non-mental attributes, including those directly related to the destination, such as expectation, quality, as well as those on the transit region/routes such as travel time, accessibility of a tourist destination [37]. Specifically, HSR, which is known as wingless aircraft, can be an important resource to attract tourists [38,39]. After the opening of HSR, the proportion of tourists within 400–1200 km (2–5 h) of the tourist destinations has been increasing [8]. Thus, HSR would exert a substantial influence on tourism flows from the tourist generating market to the destination with its distinct circle structure [40,41] and form economic circles with a radius of hours.

According to the Outline of Railway Advance Planning for China's Powerhouse in the Railway Sector in the New Era, 1-, 2- and 3-h HSR travel circles will be formed nationwide by 2035. By HSR, major urban areas (suburbs) can be reached in one hour, major cities within the city cluster can be connected within 2 h, while the adjacent urban agglomeration and provincial capital cities are accessible in 3 h. In particular, commuter flow circles within one hour have been/ will be formed in the Beijing-Tianjin-Hebei region between Beijing, Tianjin, and Xiongan, the Yangtze River Delta region between Shanghai and Suzhou, Wuxi, and Changzhou, the Guangdong-Hong Kong-Macao Greater Bay Area between Guangzhou-Shenzhen and Guangzhou-Zhuhai, and the Chengdu-Chongqing economic circle between Chengdu and Chongqing. Urban agglomeration fast channels within 2 h have been/will be formed in the Beijing-Tianjin-Hebei region between Beijing and Shijiazhuang, the Yangtze River Delta region between Shanghai, Nanjing, and Hangzhou, the Guangdong-Hong Kong-Macao Greater Bay Area between Guangzhou, Shenzhen, Hong Kong, Macao, and surrounding cities in the Pearl River Delta, and the Chengdu-Chongqing Economic circle between the surrounding cities and Chengdu/Chongqing. The capacity of main HSR

channels has been/ will be strengthened to achieve 3-h access by building the backbone of the comprehensive transportation network of the urban agglomeration.

Tourists often face time constraints on their trips and want to travel in a more environmentally friendly way [42], which mean that there is always only a limited amount of time in which people can travel for touristic activities. Tourists' mobility is therefore confined by a space-time prism [37]. In a simple word, where a tourist may take a trip to is determined by how much time he/she has and how far the destination is. To be noted, destination choice does not solely depend on the physical distance. A tourist destination may be physically close to a tourist generating region but takes a substantial amount of time in transit, including both time spent on the route and time spent waiting, checking in, at border control, etc. In some scenarios, a destination may be simply inaccessible through public transportation. From this argument, tourists' decision making in destination choices does not solely rely upon the quality of tourism products, but starts with travel time, costs, and accessibility, which fundamentally shifts the focus onto time and space.

## 3. Methodology and Data

### 3.1. Methodology

#### 3.1.1. Time-Space Compression Model

From the perspective of spatial relation theory and Leiper's tourism system theory [43], the TSC effect of HSR on tourist destination is generated by the change of relational space between tourist destination and tourist generating region, which are mainly reflected in two aspects. First, TSC effect of HSR on tourist destinations improves the accessibility of tourist destinations in remote areas, tourists can choose new spaces to meet their diversified needs. It is similar to the process of "spatial fix" [22], which is described as SE in this paper. Secondly, the changes of relational space encourage the repeat consumption of attractive and distinctive tourist attractions in the original destination. It is similar to the process of "time-space convergence" [21], which is defined as TC in this paper.

Due to its higher speed, HSR in general reduces distances in temporal sense, expands spatial space in tourism. The reform of transportation mode can change the accessibility of tourist channels, shorten the temporal distance between tourist destination and tourist generating region, while the changes of relational space have an impact on tourists, tourism and the whole tourism system. Therefore, this paper defines the TSC effect of HSR as the process in which the original consumption space of tourists is folded and the new consumption space is created under the condition that science and technology change the relational space between tourist destination and tourist generating region. In this process, the market structure of tourist destination is broken, and the travel behavior and mode of tourists are changed. In destination development, this process can be measured by changes in the tourist generating region. The TSC effect of HSR on tourist destination can be summarized according to the effects of (1) TC effect; (2) SE effect; and (3) both as TSC effect. Its theoretical framework is shown as follows (see Figure 1 for details).

In this proposed model, tourists, tourist generating region, tourist destination region, and transit channels are consistent with literature such as Hall's [37] model and Leiper's [43] tourism system theory. TC can be simplified to time reduction in travelling due to the introduction of HSR system between tourist generating region and tourist destination region. SE, on the other hand, can be simplified to the linkage between tourist generating region and tourist destination region via HSR system, which was not accessible by a certain public transportation mode (such as trains or flights). SE and TC are two important dimensions of TSC.

The calculation of the TSC effect of HSR on tourist destinations is divided into two parts. Firstly, based on the concept of 1-, 2- and 3-h HSR travel circle, considering the relatively fixed transfer time between stations and tourist destinations, we added 2 h to the temporal distance for the convenience of calculation. The basic accessibility of tourist destinations under temporal distance of 3 h, 4 h and 5 h is calculated based on the accessibility model. Basic accessibility includes airline accessibility, highway accessibility,

and ordinary train accessibility. The accessibility of HSR is the TSC effect of HSR on tourist destinations to be estimated in this paper. Then comes the estimation of the TC effect and SE effect of the HSR on tourist destinations. Although the first step estimates the TSC effect of HSR on tourist destinations, it cannot estimate the TC effect and SE effect of HSR on tourist destinations. There is an overlap between the basic accessibility and the HSR accessibility under the temporal distance of 3 h, 4 h, and 5 h. The overlapping part is the TC effect of HSR on tourist destination. Thus, we calculate the overlapping part to separate the TC effect and SE effect of HSR on the tourist destination.

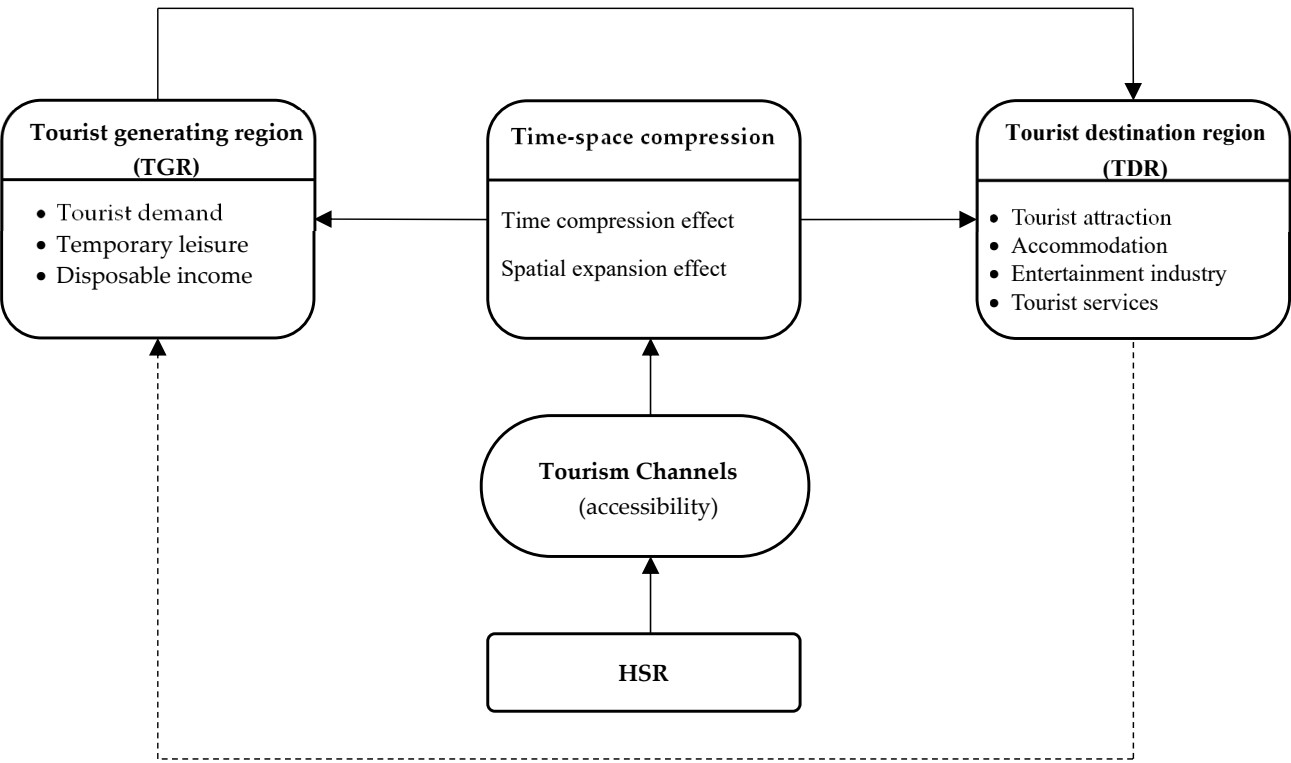

**Figure 1.** Time-space compression model in tourism context.

Accessibility of Tourist Destination

The temporal distance (e.g., 5 h) from the tourist generating region to tourist destination is divided into two sections. They are the temporal distance between the tourist destination and the transport station (airport, expressway exit, train station, HSR station) and the temporal distance between the transport station (airport, expressway exit, train station, HSR station) and the prefecture-level city respectively. The specific formula is as follows:

$$0 < T_{i,j,k} \leq 5, \theta_{i,j,k} = 1, \text{ when } T_{i,j,k} > 5, \text{ or } T_{i,j,k} = 0, \theta_{i,j,k} = 0. \tag{1}$$

TSC Effect of HSR on Tourist Destination

Although the above accessibility calculation model can estimate the accessibility of tourist destinations by planes, HSR, regular trains, and automobiles (highways), the TC effect and SE effect cannot be separated. In order to solve this problem, this paper puts forward a calculation model for TSC effect of HSR on tourist destinations. The calculation formula is as follows:

$$TSC_i = S_{i,h} \ SE_i = \sum_{j,i \neq j}^{n} G_j I \ TC_i = TSC_i - SE_i \tag{2}$$

where $TSC_i$ donates the TSC effect of HSR on tourist destination $i$ within a temporal distance of 5 h. $S_{i,h}$ is the accessibility of tourist destination $i$ by HSR within 5 h, and the unit is 10,000 people; see Formula (1) for details. $G_j$ refers to the population of the generating region $j$, whose unit is 10,000 people.

$IE_i$ is the SE effect of HSR on tourist destination $i$ within a temporal distance of 5 h. $\omega_{i,j,h}$ refers to the connection coefficient between tourist destination $i$ and tourist generating region $j$ by HSR, it's either 0 or 1. When $0 < T_{i,jIh} \leq 5, T_{I,j,a}$, I, $T_{i,j,t}$ are greater than 5 or equal to 0, then $\omega_{i,j,h} = 1$. In other cases, $\omega_{i,jIh} = 0$.

$TC_i$ donates the TC effect of HSR on tourist destination $i$.

3.1.2. Econometric Model

We use a panel model to evaluate the TSC effects of HSR on tourist destinations. The panel regression model is specified as follows:

$$TD_{it} = \alpha TSC_{it} + \varepsilon_{it} TD_{it} = \beta TC_{it} + \delta SE_{it} + \varepsilon_{it} \tag{3}$$

where $TD_{it}$ is the development index of tourist destination $i$ in year $t$. We use the nighttime light data of the destination to represent the tourism development index, i.e., a buffer zone with the tourist destination as its center, with a radius of 5 km is established, and the sum of the nighttime light data therein is calculated. $TSC_i$ represents TSC effect of HSR on tourist destination $i$ in year $t$. $TC_i$ is the TC effect of HSR on tourist destination $i$ in year $t$. $SE_i$ donates the SE effect of HSR on tourist destination $i$ in year $t$. The unit for $TSI$, $TC_i$, and $SE_i$ is 10,000 people. $\alpha$, $\beta$, and $\delta$ are the impact coefficients of the TSC effect, the TC effect, and the SE effect of HSR on the development of tourist destinations. $\varepsilon_{it}$ denotes the residual item.

*3.2. Variables and Data*

3.2.1. Variables

This paper selects the development index of tourist destination (TD) as the dependent variable. The Night Light Development Index (NLDI) is selected to reflect the development of tourist destinations. NLDI is "a simple, objective, spatially explicit and globally available empirical measurement of human development derived solely from nighttime satellite imagery and population density" [44]. NLDI, therefore, has widely been applied to study the spatial distribution of gross domestic product (GDP), reflecting the status of local economy and vitality [45–47]. To measure the human development of a target tourist destination as well as its adjacent regions, this study considered areas with each destination as the center and a radius of 5 km as units for gathering nighttime light data. Since all these locations are AAAA and AAAAA classified tourist destinations, their local economy and vitality are significantly tourism oriented, NLDI is used to estimate tourism development within these destinations. The accessibility of HSR is the TSC effect of HSR on tourist destinations to be estimated in this paper. The overlap between the basic accessibility and the HSR accessibility under temporal distance of 3 h, 4 h, and 5 h represents TC effect. In addition to TC effect, all that remains is the SE effect. The independent variables are TSC effect, TC effect, and SE effect.

3.2.2. Data Selection

This paper selects 2662 tourist destinations in China as the research samples. These destinations are classified as AAAA and AAAAA, according to the standard of Administrative Measures for the Quality Grade Evaluation of Tourist Areas carried out by the China National Tourism Administration (CNTA) in 2005 and modified in 2012. In spite of its termination in 2016, these measures provide a clear reference in classifying tourist destinations in terms of market size. For instance, a destination is graded as AAAA only if it caters to at least 500,000 tourism visits (including at least 30,000 from overseas). Accordingly, AAAA tourist destinations (as well as AAAAA graded with even higher number of visits) are included in this study to assure the emphasis on popular tourist destinations

in China. Further, 4A and 5A scenic spots in China are mainly sightseeing scenic spots. The consumption behavior of tourists in sightseeing scenic spots is different from that of vacationing scenic spots.

The data used in this study mainly come from China City Statistical Yearbook, Provincial Statistical Yearbooks, and Urban Statistical Bulletins. The distribution of the research samples is illustrated in Figure 2.

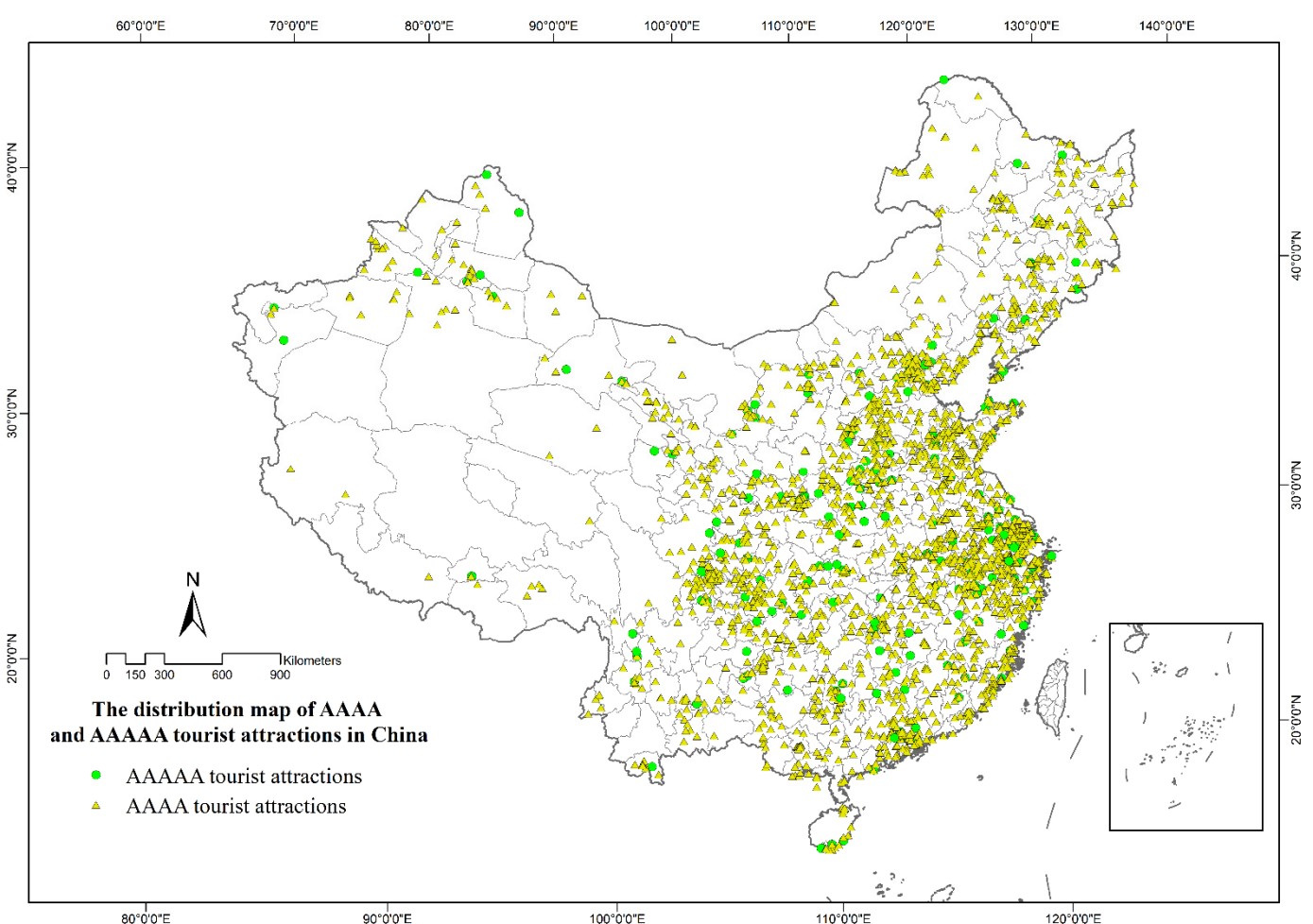

**Figure 2.** Sample distribution map.

This paper selects annual nighttime light data from 2008 to 2019. Defense Meteorological Satellite Program Operational Linescan System (DMSP-OLS) was only updated to 2013. Although the Visible Infrared Imaging Radiometer Suite (Suomi NPP Satellite, NPP/VIIRS) data from 2012 to now can be obtained, the data were produced by different satellites, resulting in inconsistencies. To avoid data inconsistencies, night light data for 2008–2019 are quoted from the dataset published by Zhong et al. [48]. We choose one part of the database named EANTLI_Like_2000–2020. (http://www.geodoi.ac.cn/WebEn/doi.aspx?Id=3100; DOI:10.3974/geodb.2022.06.01.V1, accessed on 10 September 2022). The spatial resolution is 1 km.

As shown in Table 1, the Jarque–Bera test demonstrates that the distribution of TSC, SE, TC, and NLDL is non-normal. In order to eliminate co-linearity, this paper runs a logarithmic processing for each variable.

**Table 1.** Statistics of the Panel.

| Scenario | Statistics | Mean | Median | Maximum | Minimum | Std. Dev. | Jarque-Bera | Observations |
|----------|-----------|------|--------|---------|---------|-----------|-------------|--------------|
| 3 h | *TSC* | 3118.17 | 2142.47 | 18,044.96 | 0.00 | 3232.37 | 6236.25 *** | 22,200 |
| | *SE* | 1264.25 | 800.20 | 13,293.21 | 0.00 | 1625.02 | 34,512.37 *** | 22,200 |
| | *TC* | 1853.92 | 1201.03 | 9603.15 | 0.00 | 2080.01 | 6675.69 *** | 22,200 |
| | *TD* | 6.29 | 3.78 | 31.43 | 0.00 | 6.47 | 6391.08 *** | 22,200 |
| 4 h | *TSC* | 5243.53 | 3759.08 | 32,524.24 | 0.00 | 4950.03 | 7569.96 *** | 23,400 |
| | *SE* | 2877.62 | 1980.90 | 24,859.88 | 0.00 | 3079.37 | 21,864.19 *** | 23,400 |
| | *TC* | 2365.91 | 1695.00 | 13,091.84 | 0.00 | 2508.46 | 4987.88 *** | 23,400 |
| | *TD* | 6.23 | 3.73 | 31.43 | 0.00 | 6.42 | 6994.53 *** | 23,400 |
| 5 h | *TSC* | 7834.25 | 5907.43 | 39,346.86 | 0.00 | 7153.65 | 6117.85 *** | 23,736 |
| | *SE* | 2643.96 | 1515.24 | 28,129.41 | 0.00 | 3290.99 | 52,697.42 *** | 23,736 |
| | *TC* | 5190.29 | 3874.17 | 27,821.81 | 0.00 | 4684.40 | 5322.63 *** | 23,736 |
| | *TD* | 6.23 | 3.73 | 31.43 | 0.00 | 6.42 | 7106.32 *** | 23,736 |

Notes: *** Indicates significance at the 1% level, and a balanced dataset during the period of 2008–2019.

## 4. Results

### 4.1. TSC Effect by HSR

The TSC effect of HSR on tourist destinations in eastern and central China is three times higher than that in western and north-eastern China. TSC is insignificant in western and north-eastern China. The maximum TSC market potentials under temporal distance 3 h, 4 h, and 5 h are, respectively, 10,509, 17,356, and 20,042 million people (see Figure 3). Specifically, TSC effect of HSR on tourist destinations is mainly manifest along Beijing-Shanghai belt, Beijing-Harbin belt, the Coastal belt (Shanghai-Fuzhou), the Yangtze River belt (Wuhan-Shanghai), and Wuhan-Guangzhou belt. In these belt areas, the impacts further show an uprising pattern along with the increase of temporal distance from 3 h to 5 h. The TSC effect of HSR on tourist destinations also spread to peripheral regions from the metropolis city cluster of Yangtze River Delta, Bohai Gulf metropolis city cluster, the middle Yangtze River city cluster, and the metropolis of central China. The visible TSC effect of HSR on tourist destinations in western China (North-western China) lies on the Xi'an-Lanzhou-Urumchi belt, due to a lack of HSR development in these regions. The Chengdu-Chongqing area, nevertheless, has formed its own cluster for reducing time-space constraints. To sum up, these results show an imbalanced market development for tourist destinations in China, making destination location become a central topic in the discussion of future tourism supply and demand, since HSR is and will remain the key public transportation mode in China.

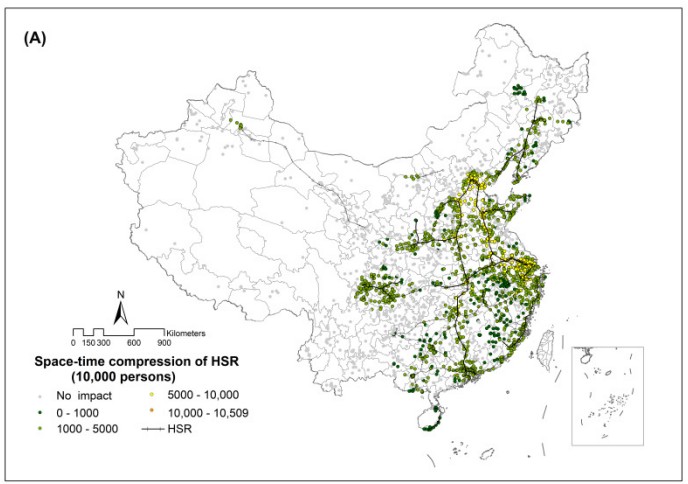
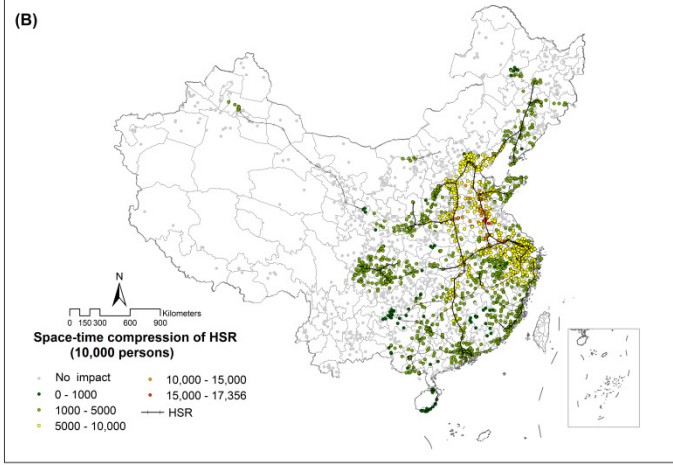

**Figure 3.** *Cont.*

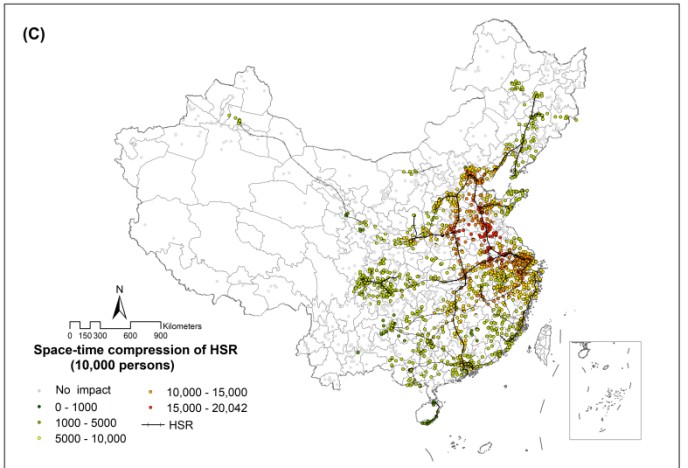

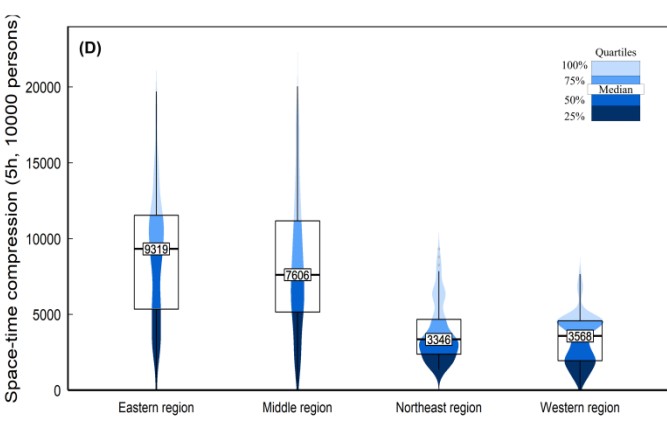

**Figure 3.** TSC effect of HSR on tourist destinations. (**A–C**) represent the TSC effect under temporal distance of 3 h, 4 h and 5 h respectively during 2008–2019. (**D**) shows the Violin Plot of TSC effect of HSR in four regions of China (5 h).

The maximum market accessibility driven by TC effect of HSR under the temporal distance of 3 h, 4 h, and 5 h is, respectively, 5461, 7955, and 14,113 million people, while the maximum market accessibility driven by the SE effect of HSR under the temporal distance of 3 h, 4 h, and 5 h is, respectively 8524, 13,810, and 14,150 million people (see Figures 4 and 5). Compared to pre-HSR era, market accessibility under short temporal distance (such as 3 h) has not increased significantly from the aspect of TC effect, in spite of rapid HSR development. However, when the temporal distance increases to 5 h, the TC effect of HSR in certain areas, such as Beijing-Shanghai belt and Shanghai-Fuzhou belt, similar to the SE effect of HSR, has fundamentally changed the market structure, by connecting many locations to a destination within a half-day distance.

### 4.2. TSC Effect of HSR on Destination Development

Table 2 reflects the results of panel unit root test by ADF–Fisher Chi-square. As can be seen from the table, for the three temporal distance panels (3 h, 4 h, 5 h), variables (lnDT, lnSTC, lnSE, and lnTC) are stationary and reject the null hypothesis whether it's at the original level or after the first difference at the significance level of 0.05. It also shows that all variables are stationary, i.e., I (1), at the significance level of 0.05. Thus, the subsequent cointegration test and panel regression are reasonable. The Pedroni cointegration test is used to identify the cointegration relationship. As shown in Table 3. For the three temporal distance panels (3 h, 4 h, 5 h), it is obvious that more statistics reject the null hypothesis (without cointegration) at the significance level of 0.05. Therefore, we conclude that there is a long-term stable relationship between the variables (lnDT, lnSTC, lnSE and lnTC), which lays a foundation for the estimation of subsequent panel regression model.

From the panel regression analysis, TC, SE, and TSC have significant negative effects on destination development (DT), in all three temporal distance scenarios, suggesting a generally negative impact of TSC effect of HSR on destination development in China. The negative impact may be seen as an inhibitory effect. The results are illustrated in Tables 4–6.

Interestingly, the effect coefficients of TSC effect of HSR on destination development manifest a descending pattern, from −0.193 in scenario 1 (3 h), to −0.117 in scenario 2 (4 h), and then to −0.091 in scenario 3 (5 h). The effects of TC and SE by HSR follow similar descending patterns. For instance, TC effect of HSR negatively affect destination development, from a coefficient of -0.196 in scenario 1 (3 h), to −0.175 in scenario 2 (4 h), and then to −0.124 in scenario 3 (5 h). The impact coefficients of TC effect of HSR on destination development within temporal distance (3 h, 4 h, and 5 h) are −0.197, −0.116, and −0.110, respectively. In another word, the farther the temporal distance, the weaker

the inhibitory effect. Considering both the TSC and SE effect of HSR, their inhibitory effects significantly reduce when the temporal distance increases from 3 to 4 h.

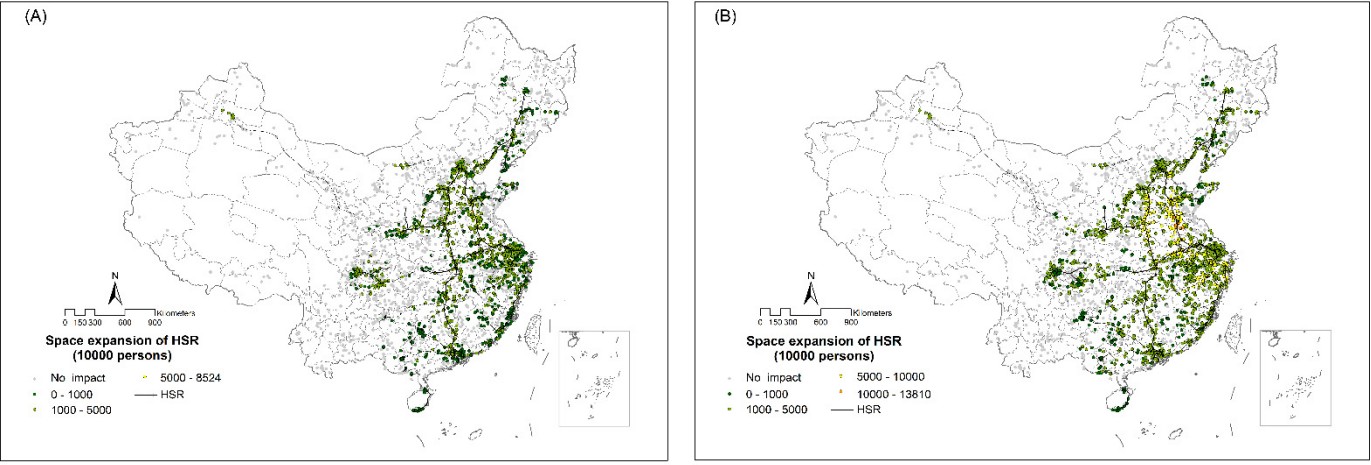

**Figure 4.** TC effect of HSR on tourist destinations. (**A–C**) represent the TC effect under temporal distance of 3 h, 4 h and 5 h respectively during 2008–2019. (**D**) shows the Violin Plot of TC effect of HSR in four regions of China (5 h).

**Figure 5.** *Cont.*

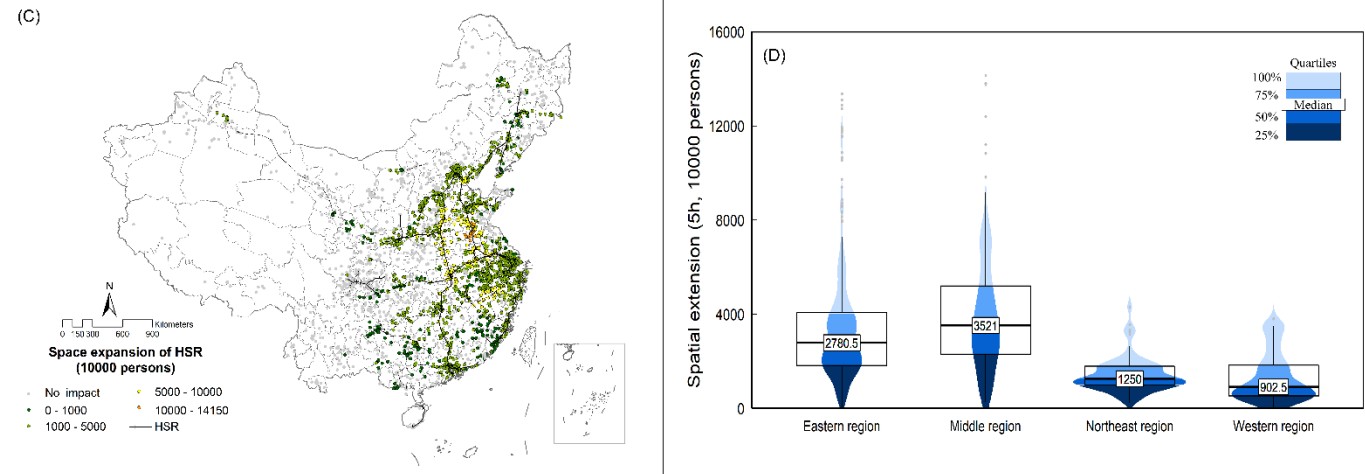

**Figure 5.** SE effect of HSR on tourist destinations. (**A–C**) represent the SE effect under temporal distance of 3 h, 4 h and 5 h respectively during 2008–2019. (**D**) shows the Violin Plot of SE effect of HSR in four regions of China (5 h).

**Table 2.** Results of panel unit root test for different scenarios.

| Variable | ADF-Fisher Chi-Square | | | |
| --- | --- | --- | --- | --- |
| | Level | | First Difference | |
| | Intercept | Intercept and Trend | Intercept | Intercept and Trend |
| | 3 h scenario | | | |
| lnDT | 9705.94 *** | 9938.05 *** | 24,482.8 *** | 18,479.2 *** |
| lnSTC | 5471.70 *** | 7883.46 *** | 16,976.4 *** | 12,788.4 *** |
| lnSE | 5111.76 *** | 5962.00 *** | 12,993.2 *** | 9253.21 *** |
| lnTC | 5140.51 *** | 6652.29 *** | 14,961.4 *** | 10,856.1 *** |
| | 4 h scenario | | | |
| lnDT | 10,277.2 *** | 10,521.3 *** | 25,826.6 *** | 19,533.2 *** |
| lnSTC | 9051.59 *** | 11,758.8 *** | 20,358.8 *** | 15,856.7 *** |
| lnSE | 8656.93 *** | 10,279.0 *** | 18,331.8 *** | 13,020.2 *** |
| lnTC | 6856.21 *** | 8538.02 *** | 16,729.3 *** | 12,625.7 *** |
| | 5 h scenario | | | |
| lnDT | 8334.73 *** | 7766.85 *** | 21,767.9 *** | 16,248.4 *** |
| lnSTC | 13,759.1 *** | 11,003.3 *** | 18,474.4 *** | 15,547.3 *** |
| lnSE | 9107.27 *** | 8840.42 *** | 15,808.2 *** | 11,801.4 *** |
| lnTC | 9360.46 *** | 7714.51 *** | 15,238.3 *** | 12,513.8 *** |

The optimal lags are selected automatically by Schwarz information criteria. Notes: *** $p < 0.01$, ** $p < 0.05$, * $p < 0.1$.

**Table 3.** Results of Pedroni cointegration test.

| Scenario | 3 h Scenario | 4 h Scenario | 5 h Scenario |
| --- | --- | --- | --- |
| Alternative hypothesis: common AR coefs. (within-dimension) | | | |
| Panel v-Statistic | −6.54 | −8.43 * | −15.43 * |
| Panel rho-Statistic | 10.98 ** | 21.29 | 29.33 |
| Panel PP-Statistic | −62.70 *** | −28.95 *** | −17.55 *** |
| Panel ADF-Statistic | 33.95 *** | 36.10 *** | 16.88 *** |
| Alternative hypothesis: individual AR coefs. (between-dimension) | | | |
| Group rho-Statistic | 31.83 | 36.76 | 39.60 |
| Group PP-Statistic | −64.31 *** | −74.07 *** | −75.12 *** |
| Group ADF-Statistic | 30.68 *** | 29.57 *** | 19.43 *** |

Notes: *** $p < 0.01$, ** $p < 0.05$, * $p < 0.1$.

**Table 4.** TSC effect of HSR on destination development (3 h).

| Explanatory | (1) | (2) | (3) | (4) |
|:---:|:---:|:---:|:---:|:---:|
| **Variables** | **lnDT** | **lnDT** | **lnDT** | **lnDT** |
| *lnTSC* | −0.193 *** | | | |
| | (0.0104) | | | |
| *lnSE* | | −0.197 *** | | −0.0763 *** |
| | | (0.0141) | | (0.0209) |
| *lnTC* | | | −0.196 *** | −0.170 *** |
| | | | (0.0119) | (0.0168) |
| *Cons* | 3.076 *** | 2.950 *** | 3.005 *** | 3.339 *** |
| | (0.0828) | (0.103) | (0.0901) | (0.126) |
| Year FE | YES | YES | YES | YES |
| TDR FE | YES | YES | YES | YES |
| Observations | 14,594 | 11,665 | 12,874 | 9945 |
| R-squared | 0.026 | 0.019 | 0.023 | 0.029 |
| F-test | 341.4 *** | 194.4 *** | 271.4 *** | 127.5 *** |

Notes: Dependent variable: *lnDT*. Standard errors in parentheses, * $p < 0.1$, ** $p < 0.05$, and *** $p < 0.01$. The estimation approach selection is based on the Hausman test.

**Table 5.** TSC effect of HSR on destination development (4 h).

| Explanatory | (1) | (2) | (3) | (4) |
|:---:|:---:|:---:|:---:|:---:|
| **Variables** | **lnDT** | **lnDT** | **lnDT** | **lnDT** |
| *lnTSC* | −0.117 *** | | | |
| | (0.00767) | | | |
| *lnSE* | | −0.116 *** | | −0.0449 *** |
| | | (0.00851) | | (0.0144) |
| *lnTC* | | | −0.175 *** | −0.152 *** |
| | | | (0.00992) | (0.0140) |
| *Cons* | 2.547 *** | 2.472 *** | 2.921 *** | 3.093 *** |
| | (0.0642) | (0.0665) | (0.0774) | (0.0913) |
| Year FE | YES | YES | YES | YES |
| TDR FE | YES | YES | YES | YES |
| Observations | 16,714 | 15,535 | 13,954 | 12775 |
| R-squared | 0.015 | 0.013 | 0.025 | 0.027 |
| F-test | 232.2 *** | 185.6 *** | 311.3 *** | 154.5 *** |

Notes: Dependent variable: lnDT. Standard errors in parentheses, * $p < 0.1$, ** $p < 0.05$, and *** $p < 0.01$. The estimation approach selection is based on the Hausman test.

Further, the TSC and SE effect of HSR on destination development show similar impacts across three temporal scenarios. However, for TC and SE effect of HSR, (1) TC effect of HSR (β = −0.170, $p < 0.01$) illustrates a relatively stronger inhibitory effect than SE effect of HSR (β = −0.076, $p < 0.01$) in scenario 1 (3 h); (2) TC effect of HSR (β = −0.152, $p < 0.01$) illustrates a significantly stronger inhibitory effect than SE effect of HSR (β = −0.045, p. < 0.01) in scenario 2 (4 h); and (3) TC effect of HSR (β = −0.059, $p < 0.01$) illustrates a relatively weaker inhibitory effect than SE effect of HSR (β = −0.096, $p < 0.01$) in scenario 1 (5 h).

### 4.3. Robustness Test

In order to further confirm the TSC effect of HSR on the development of tourist destinations, and to eliminate the influence of other factors, this paper introduces tourist destination Baidu search index as a control variable. The higher the tourist destination Baidu search index, the greater the potential of tourist generating region and the greater the influence of the tourist destination. To a certain extent, the introduction of tourist destination in Baidu search index can eliminate the difference in the impact of tourist generating region on the development of tourist destinations. Since the historical Baidu search index can only be obtained from 2017, this paper chooses 2017, 2018, and 2019 as the

verification period, and the tourist destination Baidu search index is based on data from websites (http://index.baidu.com/v2/index.html#/, accessed on 10 September 2022).

**Table 6.** TSC effect of HSR on destination development (5 h).

| Explanatory | (1) | (2) | (3) | (4) |
|---|---|---|---|---|
| Variables | lnDT | lnDT | lnDT | lnDT |
| *lnTSC* | −0.0910 *** | | | |
| | (0.00651) | | | |
| *lnSE* | | −0.110 *** | | −0.0958 *** |
| | | (0.00828) | | (0.0143) |
| *lnTC* | | | −0.124 *** | −0.0588 *** |
| | | | (0.00771) | (0.0149) |
| *Cons* | 2.358 *** | 2.412 *** | 2.607 *** | 2.801 *** |
| | (0.0566) | (0.0650) | (0.0645) | (0.0850) |
| Year FE | YES | YES | YES | YES |
| TDR FE | YES | YES | YES | YES |
| Observations | 17,815 | 13,952 | 17,171 | 13,308 |
| R-squared | 0.012 | 0.014 | 0.017 | 0.019 |
| F-test | 195.3 *** | 178.1 *** | 260.6 *** | 116.9 *** |

Notes: Dependent variable: lnDT. Standard errors in parentheses, * $p < 0.1$, ** $p < 0.05$, and *** $p < 0.01$. The estimation approach selection is based on the Hausman test.

In this paper, 2662 tourist destinations are selected as research samples, but only 887 of them are included in Baidu search index. Therefore, this paper selects these 887 samples for verification analysis. As shown in Table 7, for the scenario 1 (3 h), the TSC effect of HSR has a negative impact on the development of tourist destinations (−0.242). After introducing the tourist destination Baidu search index, the TSC effect of HSR still has a negative impact on the development of tourist destinations, but its influence drops by 9%, which partially eliminates the impact of market size. For scenario 2 and 3 (4 h, 5 h), the TSC effect of HSR has a negative impact on the development of tourist destinations (−0.142 and −0.116, respectively). After introducing the tourist destination Baidu search index, the TSC effect of HSR still has a negative impact on the development of tourist destinations, but its influence drops by less than 0.5%, which partially demonstrates that the size of the destination market matters less for the temporal distance (4 h, 5 h). From the above conclusion, it can be concluded that after introducing the tourist destination Baidu search index, the negative impact of the TSC effect of HSR on the development of tourist destinations has not changed. Moreover, the longer the temporal distance, the smaller the negative influence, which basically confirms the rationality of the research conclusion in this paper.

**Table 7.** Robustness test.

| Explanatory | lnnit (3 h) | | lnnit (4 h) | | lnnit (5 h) | |
|---|---|---|---|---|---|---|
| Variables | Testing Model | Original Model | Testing Model | Original Model | Testing Model | Original Model |
| lnIndex | 0.0383 ** | | 0.0337 ** | | 0.0336 ** | |
| | (0.0166) | | (0.0160) | | (0.0161) | |
| lnSTC | −0.152 *** | −0.242 *** | −0.145 *** | −0.142 *** | −0.123 *** | −0.116 *** |
| | (0.0329) | (0.0143) | (0.0344) | (0.0096) | (0.0335) | (0.0079) |
| _cons | 2.226 *** | 3.368 *** | 2.311 *** | 2.675 *** | 2.172 *** | 2.497 *** |
| | (0.260) | (0.112) | (0.281) | (0.0792) | (0.283) | (0.0671) |
| Year FE | YES | YES | YES | YES | YES | YES |
| TDR FE | YES | YES | YES | YES | YES | YES |

**Table 7.** *Cont.*

| Explanatory | lnnit (3 h) | | lnnit (4 h) | | lnnit (5 h) | |
|---|---|---|---|---|---|---|
| Variables | Testing Model | Original Model | Testing Model | Original Model | Testing Model | Original Model |
| Observations | 2235 | 7700 | 2273 | 9067 | 2298 | 9807 |
| R-squared | 0.0233 | 0.046 | 0.0205 | 0.029 | 0.0191 | 0.027 |
| F-test | 22.56 | 286.4 *** | 18.43 | 218.0 *** | 14.25 | 218.3 *** |

Notes: Standard errors in parentheses, * $p < 0.1$, ** $p < 0.05$, and *** $p < 0.01$. The estimation approach selection is based on the Hausman test. lnnit represents the tourist destination development index, Lnindex represents tourist destinations Baidu search index, lnstc represents the TSC effect of HSR on tourist destinations.

## 5. Discussion

Results on TSC effect of HSR suggest an imbalanced domestic tourism market structure in China. Tourist destinations in eastern and central regions of China have accesses to a wider and more diversified market, thanks to HSR development in recent years. Similar patterns are found between temporal distances (4 h) and temporal distances (5 h) in the TSC effect of HSR. In particular, many tourist destinations along Beijing-Shenyang belt, Beijing-Xi'an belt, and Beijing-Jinan belt have access to a population of over 100 million, illustrating the key tourism markets in North China. Other mega regional markets are found in eastern China, in the metropolis city cluster of Yangtze River Delta; Coastal belt (Shanghai-Fuzhou); South China, centering the Guangdong-Hong Kong-Macao Greater Bay Area; and Chengdu-Chongqing cluster. Several provinces with relatively large number of AAAA and AAAAA tourist destinations are not supported by markets with corresponding sizes, such as Heilongjiang, Yunnan, and Guizhou Provinces.

The TC effect of HSR is not significant but with similar patterns at temporal distances (3 and 4 h), whereas at 5 h temporal distance, TC effect of HSR is evident. With the development of HSR, many tourist destinations along with Beijing-Shanghai belt and Yangtze River Delta area are now able to reach more than 50 million residents previously not accessible within five hours. HSR has doubtless brought many tourist markets closer together in China in general. As for the SE effect of HSR, Central regions in China, specifically, were found with the most significant SE effect of HSR, in contrast to others. A latitudinal pattern of these regions' locations is found along the Yellow River and Yangtze River. Another pattern of distribution of these regions is along the eastern coast of China. A second altitudinal pattern is found along the famous Beijing Kowloon Railway, meeting Yangtze River pattern at Wuhan. In contrast to the TC effect of HSR that brings tourist markets close, the SE effect of HSR has an even greater impact on many tourist destinations in China, enabling accesses to many markets that were simply unavailable.

Compared to pre-HSR era, short temporal distance market accessibility has not risen significantly based on TC effect of HSR, suggesting a limited function of HSR on reaching nearby tourist markets. As discussed earlier, from a time perspective, time saved by taking HSR is marginal, considering the same (sometimes increased) time spent on transitions from home to the train station as well as from the train station to accommodation. From a cost perspective, time saved by taking HSR is relatively not comparable in value, considering the significantly increased price purchasing HSR tickets.

Considering short temporal distance markets, HSR may bring in more challenges rather than opportunities. This easy access to more adjacent attractions exposes many tourist destinations from competing with local destinations, to competing against a bigger number of competitors with stronger competitive advantages and greater reputations in a larger geographical region. Smaller and emerging destinations close to one mega destination city may enjoy a spill-over effect, but many close to two and more than two metropolises may, on the other hand, bear a backwash effect instead. For domestic tourism in China, the longest holiday is seven days, twice a year. This wallet of time may be solely shared by two adjacent metropolises, such as Beijing and Tianjin, Shanghai and Suzhou, Shenzhen and Hong Kong, and leave small tourism cities no chance to host many tourists.

Therefore, destination management organizations (DMOs) need to carefully design their tourism products to better attract tourists available via the introduction of HSR and compete against more domestic tourist destinations. DMOs in mega cities may enjoy siphon effect [49], while DMOs in small destinations need to develop unique selling points to neutralize any backwash effect, or even encourage spill-over from near mega tourist destinations.

At short temporal distance, TSC effect of HSR is found an interesting negative impact on regional development for tourist destinations. Besides general backwash effects by HSR on smaller cities, this inhibitory effect may be explained with regard to other contexts. From a marketing point of view, the tourism development of destinations may have not caught up with the increased market size enabled by HSR.

Furthermore, although HSR can increase tourists' travel frequency and expand the travel consumption radius, it also significantly reduces the number of overnight tourists, especially the number of weekend leisure tourists. This, to a certain extent, inhibits the development of the hotel industry around the tourist attractions. In addition, HSR passengers are often very sensitive with regard to travel time, and the main motivation for choosing HSR is to shorten travel time. For instance, Wang et al. [50] ran a survey involving more than 5000 HSR passengers at Beijing, Nanjing, and Tianjin HSR stations, finding that for 70% of passengers, shorter travel time was key in choosing HSR. Tourists therefore value HSR in significantly increasing physical distance within the same duration and thus prefer to travel to farther destinations rather than temporally close ones [51].

In spite of the inhibitory effect, close-range HSR tourist generating market is not necessarily unimportant for tourist destinations. On the contrary, it indicates that the HSR short-haul tourist generating market of tourist destinations has shortened the cycle of production, exchange, and consumption, and has fundamentally changed the nature of demand and consumption due to the shortening of time and distance. Therefore, the development of tourist destinations needs to rely on resource advantages and regional advantages to accurately meet the market demand of HSR tourist generating market as well as upgrade and improve the tourist product structure of destinations.

## 6. Conclusions

Time-compression theory by Harvey [22,23] and many other scholars' following work were discussed to develop a TSC model in the tourism context in relation to HSR. The TSC effect of HSR on destination development was examined based on the TSC model. Different temporal distances are distinguished in understanding the TSC effect of HSR and its impact on destination development.

Due to the time-saving convenience of HSR, the laying of the HSR network, and the auxiliary connectivity of other transportation facilities, the diversion effect of HSR on tourists is unusually obvious. The appearance of HSR not only changes tourists' perception of psychological distance to destinations, but also changes the scope of a destination's appeal. As for tourists, the technological development in transportation enhances their capabilities in reaching multiple tourist destinations within a single trip, and planning and executing an enjoyable travel plan with time and space constraints. Their travel radius is expanded by the TSC effect, they gradually modify their perceptions of space and time, and fundamentally change their tourism consumption behaviors in the end. The reshaping of the tourist market will finally restructure the tourism market. As for the tourism market, the distribution pattern of the medium- and long-distance domestic tourism market is changed. With the network pattern formed by HSR channels and urban agglomeration, the TSC effect gradually expands outward. In addition, a negative relation is found between the TSC effect of HSR and destination (or regional) development. Generally speaking, HSR in China has fundamentally changed the tourist generating market structure, showing certain geographical patterns.

For tourism development, more attention should be paid to the collaborations of geographical regions which include multiple destinations. As the research shows, the TSC

effect of HSR on tourist destinations in eastern and central China is three times stronger than that in western and north-eastern China, and the farther the temporal distance, the weaker the inhibitory effect. Thus, transportation policies and the layout of the HSR network should be integrated into tourism planning, especially for those tourist transit areas. Meanwhile, in the HSR era, the destination brand positioning, selection of target customer groups, and tourism marketing in urban agglomerations should be taken into account. Policymakers should have an explicit focus on the seamless connection between tourist destinations and HSR stations, and develop an integrated, efficient transport system that is sustainable to solve potential traffic problems.

**Author Contributions:** Conceptualization, Taohong Li; methodology, Hong Shi and Ning Chris Chen; software, Hong Shi; formal analysis, Taohong Li; writing—original draft preparation, Taohong Li; writing—review and editing, Hong Shi and Ning Chris Chen; supervision, Hong Shi and Ning Chris Chen. Luo Yang rechecked the latest statistics. All authors have read and agreed to the published version of the manuscript.

**Funding:** This research was funded by the Southwest Minzu University Research Startup Funds, grant number RQD2021038.

**Institutional Review Board Statement:** Not applicable.

**Informed Consent Statement:** Not applicable.

**Data Availability Statement:** Some or all data and models that support the findings of this study are available from the corresponding author upon reasonable request.

**Conflicts of Interest:** The authors declare no conflict of interest.

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
