# Peer review of "Time-Space Compression Effect of High-Speed Rail on Tourist Destinations in China"

_ijgi, doi:10.3390/ijgi11100528_

Round 1

Reviewer 1 Report

This article has the following problems:

(1)The abstract stated that the research period was 2008-2013, but the text became 2008-2019. Please clarify the design of the full text.

(2)The accuracy of the full text data is questionable. The article shows that the night light data is DMSP/OLS. If the research period of the article is really 2003-2019, the night light data of 2014-2019 is NPP/VIIRS. As we all know, DMSP/OLS data is only updated to 2013.

(3)Should the authors include control variables in the panel model? From the current article, the R-Square value of the model is too small

(4)The reference is too old, the author needs to update the recent literature

Author Response

This article has the following problems

(1)The abstract stated that the research period was 2008-2013, but the text became 2008-2019. Please clarify the design of the full text.

Response: Line 13 on page 1. Thanks for the comments. We accepted the suggestions, and correct the research period (2008-2019) in the abstract.

(2)The accuracy of the full text data is questionable. The article shows that the night light data is DMSP/OLS. If the research period of the article is really 2003-2019, the night light data of 2014-2019 is NPP/VIIRS. As we all know, DMSP/OLS data is only updated to 2013.

Response: Line 368-377 on page 9. Thanks for the comments. We didn't make it clear where the data came from. Our research period was 2008-2019, the nighttime light data from 2008 to 2013 are based on DMSP/OLS data. The data for 2014-2019 was quoted from the data base published by Zhong et al. Based on DMSP-OLS dataset, after data preprocessing, correction and data fusion, they obtained China's long time series nighttime light dataset (2000-2020), (http://www.geodoi.ac.cn/WebEn/doi.aspx?Id=3100; DOI:10.3974/geodb.2022.06.01.V1. We have modified the data presentation in the full text.

“Although the Visible Infrared Imaging Radiometer Suite (Suomi NPP Satellite, NPP/VIIRS) data from 2012 to now can be obtained, there are in-consistencies in the nighttime light data produced by different satellites. Thus, the nighttime light data from 2008 to 2013 are based on DMSP/OLS data. The data for 2014-2019 was quoted from the data base published by Zhong et al. [49]. Based on DMSP-OLS dataset, after data preprocessing, correction and data fusion, they obtained China's long time series nighttime light dataset (2000-2020), (http://www.geodoi.ac.cn/WebEn/doi.aspx?Id=3100; DOI:10.3974/geodb.2022.06.01.V1.). The consistency of the two data is guaranteed, making the study credible.”

(3)Should the authors include control variables in the panel model? From the current article, the R-Square value of the model is too small.

Response: Thanks for the comments.

For the question of control variables. The object of our study is 2662 scenic spots in China. It is difficult to select the continuous panel data of each scenic spot from 2008 to 2019, so no control variables were used. But in order to make the research reliable, this article has carried out an Robustness test. The historical Baidu search index of 2662 scenic spots in China were used to replace the development index of tourist destination to further confirm the reliability of the study.

For the problem question of R-Square value of the model with R2 being too low. Generally speaking, when studying the data at the micro level, R-Square value may be very low or even lower than 0.1, and R2 is an indicator to measure the predictive ability of a model. Generally speaking, R2 is not used as a decisive factor in panel regression that focuses on causal identification, but needs to meet the characteristics of data consistency.

In this paper, panel unit root test by ADF -- Fisher Chi-Square and Pedroni cointegration test were used to test the stationary of data, and the results showed good reliability.

“Table 2 reflects the results of panel unit root test by ADF–Fisher Chi-square. As can be seen from the table, for the three temporal distance panels (3 hours, 4 hours, 5 hours), variables (lnDT,lnSTC,lnSE and lnTC) are stationary and reject the null hypothesis whether it's at the original level or after the first difference at the significance level of 0.05. It also shows that all variables are stationary, i.e. I (1), at the significance level of 0.05. Thus, the subsequent cointegration test and panel regression are reasonable. The Pedroni cointegration test is used to identify the cointegration relationship. As shown in Table 3. For the three temporal distance panels (3 hours, 4hours, 5 hours), it is obvious that more statistics reject the null hypothesis (without cointegration) at the significance level of 0.05. Therefore, we conclude that there is a long-term stable relationship between the variables (lnDT, lnSTC, lnSE and lnTC), which lays a foundation for the estimation of subsequent panel regression model.”

(4)The reference is too old, the author needs to update the recent literature

Response: Line 45-47 on page 1-2, line 116 on page 3 and line 189 on page 4. Thanks for the comments. We accepted the suggestions, and updated the recent literatures. References 5-9, 25-26, 37, 42 are the updated documents.

Reviewer 2 Report

This study evaluates the time-space compression effect of HSR on a sample of 2662 classified tourist destinations in China with the help of GIS technology. The study is novel and well structured, but I still have the following questions in this study:

1、Pay attention to the details. The first occurrence of the acronym HSR in the abstract is not explained. When the acronym is used again (3.1), no corresponding explanation is required.

2. What is the significance of the research in this study, and what is the theoretical value and application value of the research? It is recommended that the author briefly explain in the abstract

3. The figures need to be standardized. Several of the pictures in this article lack the scale of the basic elements of the map. The pictures in Figures 3, 4 and 5 are not clear enough. It is recommended that the author check whether the resolution of the figures meets the standard, or change the size of the figures to better presented to readers. In addition, in Figure 2 Sample distribution map, it is recommended to label the latitude and longitude of China outside the frame.

4、Why did the author set the calculation time as 3, 4, and 5 hours in the accessibility model? What is the basis for the setting? It is necessary to demonstrate its rationality.

5、What is the time scale of this article? The time dimension in the abstract is 2008-2013, and the following text indicates that the time dimension is 2008-2019?

6、Why is the data in recent years not used? All the data are dynamically updated. Have you considered whether this impact will affect the conclusion of this paper?

The influence of high-speed rail on ice–snow tourism in northeastern China. Tourism Management(2020), doi:10.1016/j.tourman.2019.104070.

Effects of rural revitalization on rural tourism. Journal of Hospitality and Tourism Management (2021),https://doi.org/10.1016/j.jhtm.2021.02.008.

He, B. J., Zhao, D. X., Zhu, J., Darko, A., & Gou, Z. H. (2018). Promoting and implementing urban sustainability in China: An integration of sustainable initiatives at different urban scales. Habitat International, 82, 83-93.

Author Response

1.Pay attention to the details. The first occurrence of the acronym HSR in the abstract is not explained. When the acronym is used again (3.1), no corresponding explanation is required.

Response: Line12 on page 1 and line 186 on page 4. Thanks for the comments. We accepted the suggestions, and explained the acronym HSR in the abstract and corrected the acronym HSR in the whole paper

Line 11 on page 1, line 84 on page 2. We also explained TSC (time-space compression effect), time compression (TC) and spatial expansion (SE) when they first appear in the paper and checked the acronym and revised them all.

  1. What is the significance of the research in this study, and what is the theoretical value and application value of the research? It is recommended that the author briefly explain in the abstract

Response: Line 11 and line 21-24 on page 1. Thanks for the comments. We accepted the suggestions, and explained the theoretical value and application value of the research in the abstract.

“This study proposes a time-space compression (TSC) model……

Findings provided managerial implications suggesting tourist destinations should implement marketing policy to retain tourists and prevent the loss of tourists brought by the opening of HSR.”

  1. The figures need to be standardized. Several of the pictures in this article lack the scale of the basic elements of the map. The pictures in Figures 3, 4 and 5 are not clear enough. It is recommended that the author check whether the resolution of the figures meets the standard, or change the size of the figures to better presented to readers. In addition, in Figure 2 Sample distribution map, it is recommended to label the latitude and longitude of China outside the frame.

Response: Thanks for the comments, we accepted the suggestions, changes were made as suggested.

  • The pictures in Figures 3, 4 and 5 are not clear enough.

We complemented the scale of the basic elements of the map, in addition, we improved the resolution of the graph from 300dpi to 1200dpi.

Figure 3. TSC effect of HSR on tourist destinations. (A), (B) and (C) represent the TSC effect under temporal distance of 3 hours, 4 hours and 5 hours respectively during 2008–2019. (D) shows the Violin Plot of TSC effect of HSR in four regions of China (5h).

Figure 4. TC effect of HSR on tourist destinations.  (A), (B) and (C) represent the TC effect under temporal distance of 3 hours, 4 hours and 5 hours respectively during 2008–2019. (D) shows the Violin Plot of TC effect of HSR in four regions of China (5h).

Figure 5. SE effect of HSR on tourist destinations. (A), (B) and (C) represent the SE effect under temporal distance of 3 hours, 4 hours and 5 hours respectively during 2008–2019. (D) shows the Violin Plot of SE effect of HSR in four regions of China (5h).

(2)“in Figure 2 Sample distribution map, it is recommended to label the latitude and longitude of China outside the frame.”

We labelled the latitude and longitude of China outside the frame.

Figure 2. Sample distribution map

4.Why did the author set the calculation time as 3, 4, and 5 hours in the accessibility model? What is the basis for the setting? It is necessary to demonstrate its rationality.

Response: Line 166-187 on page 4 and line 246-249 on page 6. Thanks for the comments. We accepted the suggestions, and explained the reason for setting the calculation time as 3, 4, and 5 hours in the accessibility model by sorting out the research conclusions on the range of tourist destinations that can be radiated by HSR and expressing the radiable range in terms of time.

“After the opening of HSR, the proportion of tourists within 400-1,200 km (2-5 hours) of the tourist destination has been increasing. Thus, it would exert a substantial influence on tourism flows from the tourist generating market to the destination with its distinct circle structure and form economic circles with a radius of hours.

According to the Outline of Railway Advance Planning for China's Powerhouse in the Railway Sector in the New Era, 1-, 2- and 3-hour HSR travel circles will be formed nationwide by 2035. By HSR, major urban areas (suburbs) can be reached in 1-hour, major cities within the city cluster can be connected within 2 hours, the adjacent urban agglomeration and provincial capital cities are accessible in 3 hours. In particular, commuter flow circles within 1 hour have been/ will be formed in the Bei-jing-Tianjin-Hebei region between Beijing, Tianjin and Xiongan, the Yangtze River Delta region between Shanghai and Suzhou, Wuxi and Changzhou, the Guangdong-Hong Kong-Macao Greater Bay Area be-tween Guangzhou-Shenzhen and Guangzhou-Zhuhai, and the Cheng-du-Chongqing economic circle between Chengdu and Chongqing. Urban agglomeration fast channels within 2 hours have been/ will be formed in the Beijing-Tianjin-Hebei region between Beijing and Shijiazhuang, the Yangtze River Delta region between Shanghai and Nanjing and Hangzhou, the Guangdong-Hong Kong-Macao Greater Bay Area between Guangzhou-Shenzhen-Hong Kong-Macao and surrounding cities in the Pearl River Delta, and the Chengdu-Chongqing Economic circle between the surrounding cities and Chengdu/Chongqing. The capacity of main HSR channels have been/ will be strengthened to achieve 3-hour access by building the backbone of the comprehensive transportation network of the urban agglomeration.”

“The calculation of time-space compression effect of HSR on tourist destinations is divided into two parts. Firstly, basing on the concept of 1-, 2- and 3-hour HSR trave circle, considering the relatively fixed transfer time between stations and tourist destinations, we added 2 hours to the temporal distance for the convenience of calculation.”

5.What is the time scale of this article? The time dimension in the abstract is 2008-2013, and the following text indicates that the time dimension is 2008-2019?

Response: Line 13 on page 1. Thanks for the comments. We accepted the suggestions, and correct the research period (2008-2019) in the abstract.

  1. Why is the data in recent years not used? All the data are dynamically updated. Have you considered whether this impact will affect the conclusion of this paper?

Response: Thanks for the comments. At present, some urban population data are incomplete in the prefecture-level city statistics of 2020 and 2021 in China. In addition, our research period is from 2008 to 2019, with a total of 12 years, which has good credibility as a panel model.

The influence of high-speed rail on ice–snow tourism in northeastern China. Tourism Management(2020), doi:10.1016/j.tourman.2019.104070.

Effects of rural revitalization on rural tourism. Journal of Hospitality and Tourism Management (2021),https://doi.org/10.1016/j.jhtm.2021.02.008.

He, B. J., Zhao, D. X., Zhu, J., Darko, A., & Gou, Z. H. (2018). Promoting and implementing urban sustainability in China: An integration of sustainable initiatives at different urban scales. Habitat International, 82, 83-93.

Response: Thanks for the recommendation, we have cited the recommended article as references 8, 42, and 7 respectively.

Reviewer 3 Report

The article addresses a relatively interesting topic, although the conclusions part should be developed more, avoiding to conclude aspects that were already known at the time of writing the article. (eg. line 613, 614)

Overall, the applied workflow is very well detailed, but in some places the need to extract the essentials is felt. The workflow could have been more schematized.

It would be good to develop some ideas to deepen the practical applicability of the study, in a clearer manner.

The resolution of the figures can be improved, with the idea of being able to graphically analyze what is happening in certain locations. Perhaps some figures would be useful to highlight the situation at a higher zoom level.

Author Response

Comments and Suggestions for Authors

The article addresses a relatively interesting topic, although the conclusions part should be developed more, avoiding to conclude aspects that were already known at the time of writing the article. (eg. line 613, 614)

Response: Line 607-638 on page 18. Thanks for the comments. We accepted the suggestions, and rewrote the conclusion part.

“Time-compression theory by Harvey[22, 23] and many other scholars’ following work were discussed to develop a TSC model in tourism context in relation to HSR. The TSC effect of HSR on destination development was examined based on the TSC model. Different temporal distances are distinguished in understanding TSC effect of HSR and its impact on destination development.

Due to the time-saving convenience of HSR, the laying of the HSR network, and the auxiliary connectivity of other transportation facilities, the diversion effect of HSR on tourists is unusually obvious. The appearance of HSR not only changes tourists' perception of psychological distance to destinations, but also changes the scope of a destination's appeal. As for tourists, the technological development in transportation enhances their capabilities in reaching multiple tourist destinations within a single trip, and planning and executing an enjoyable travel plan with time and space constraints. Their travel radius is expanded by the TSC effect, they gradually modify their perceptions of space and time, and fundamentally change their tourism consumption behaviors in the end. The reshaping of the tourist market will finally restructure the tourism market. As for tourism market, the distribution pattern of medium and long-distance domestic tourism market is changed. With the network pattern formed by HSR channel and urban agglomeration, the TSC effect gradually expands outward. In addition, a negative relation is found between TSC effect of HSR and destination (or regional) development. Generally speaking, HSR in China has fundamentally changed tourist generating market structure, and shows certain geographical patterns.

For tourism development, more attention should be paid to the collaborations of geo-graphical regions which include multiple destinations. As the research shows that the TSC effect of HSR on tourist destinations in eastern and central China is three times stronger than that in western and north-eastern China, and the farther the temporal distance is, the weaker the inhibitory effect is. Thus, transportation policies and the layout of the HSR network should be integrated into tourism planning, especially for those tourist transit areas. Meanwhile, in the HSR era, the destination brand positioning, selection of target customer groups, and tourism marketing in urban agglomeration should be taken into account. Policymakers should have an explicit focus on the seamless connection be-tween tourist destinations and HSR stations, and develop an integrated, efficient transport system that is sustainable to solve potential traffic problem.”

Overall, the applied workflow is very well detailed, but in some places the need to extract the essentials is felt. The workflow could have been more schematized.

Response: Line 205 on page 5. Thanks for the comments. We accepted the suggestions, and changed “3.1.1 Time-space compression of high-speed rail on tourist destination” into “3.1.1 Time-space compression model” to make the list of the paper clearer.

It would be good to develop some ideas to deepen the practical applicability of the study, in a clearer manner.

Response: Line 11 and line 21-24 on page 1. Thanks for the comments. We accepted the suggestions, and explained the theoretical value and application value of the research in the abstract.

“This study proposes a time-space compression (TSC) model……

Findings provided managerial implications suggesting tourist destinations should implement marketing policy to retain tourists and prevent the loss of tourists brought by the opening of HSR.”

The resolution of the figures can be improved, with the idea of being able to graphically analyze what is happening in certain locations. Perhaps some figures would be useful to highlight the situation at a higher zoom level.

Response: Thanks for the comments. We accepted the suggestions, and we improved the resolution of the graph from 300dpi to 1200dpi.

Figure 3. TSC effect of HSR on tourist destinations. (A), (B) and (C) represent the TSC effect under temporal distance of 3 hours, 4 hours and 5 hours respectively during 2008–2019. (D) shows the Violin Plot of TSC effect of HSR in four regions of China (5h).

Figure 4. TC effect of HSR on tourist destinations.  (A), (B) and (C) represent the TC effect under temporal distance of 3 hours, 4 hours and 5 hours respectively during 2008–2019. (D) shows the Violin Plot of TC effect of HSR in four regions of China (5h).

Figure 5. SE effect of HSR on tourist destinations. (A), (B) and (C) represent the SE effect under temporal distance of 3 hours, 4 hours and 5 hours respectively during 2008–2019. (D) shows the Violin Plot of SE effect of HSR in four regions of China (5h).

Round 2

Reviewer 1 Report

The author still has not solved the data problem.The author worte in the paper:The data for 2014-2019 was based on DMSP-OLS dataset.But the dataset of Zhong is the a long time series nighttime light dataset of China (2000-2020) using data pretreatment, data correction and data fusion from annually DMSP-OLS (2000-2013) nighttime light data and monthly NPP-VIIRS (April 2012-December 2020) nighttime light data.That is to say, Zhong's 2012-2020 dataset is based on NPP/VIIRS rather than DMSP-OLS dataset.Please the author seriously revise this question.Otherwise I would not agree to the publication of this paper.

Author Response

This article has the following problems

The author still has not solved the data problem. The author worte in the paper: The data for 2014-2019 was based on DMSP-OLS dataset. But the dataset of Zhong is the a long time series nighttime light dataset of China (2000-2020) using data pretreatment, data correction and data fusion from annually DMSP-OLS (2000-2013) nighttime light data and monthly NPP-VIIRS (April 2012-December 2020) nighttime light data. That is to say, Zhong's 2012-2020 dataset is based on NPP/VIIRS rather than DMSP-OLS dataset. Please the author seriously revise this question. Otherwise I would not agree to the publication of this paper.

Response: Line 18 on page 1, line 356-362 on page 8 and table1-6. Thanks for the comments. The two data base we selected before do have the problem of inconsistent caused by different satellites. To solve this problem, night light data for 2008-2019 is quoted from the dataset published by Zhong et al. We choose one part of the database which named EANTLI_Like_2000-2020. (http://www.geodoi.ac.cn/WebEn/doi.aspx?Id=3100;DOI:10.3974/geodb.2022.06.01.V1.). The spatial resolution is 1 km. We reran the experiment and the analysis; results are shown in Table 1-6. Compared with the previous research results, there are some changes in value.

Line 18 on page 1: (2) the negative impact coefficient of TSC of HSR on tourist destination development in China within temporal distances (3 hours, 4 hours and 5 hours) are -0.193, -0.117 and -0.091 respectively;

Line 356-362 on page 8: This paper selects annual nighttime light data from 2008 to 2019. Defense Meteorological Satellite Program Operational Linescan System (DMSP-OLS) was only updated to 2013. Although the Visible Infrared Imaging Radiometer Suite (Suomi NPP Satellite, NPP/VIIRS) data from 2012 to now can be obtained, the data was produced by different satellites, resulting in inconsistencies. To avoid data inconsistencies, night light data for 2008-2019 is quoted from the dataset published by Zhong et al.[48]. We choose one part of the database which named EANTLI_Like_2000-2020. (http://www.geodoi.ac.cn/WebEn/doi.aspx?Id=3100; DOI:10.3974/geodb.2022.06.01.V1.). The spatial resolution is 1 km.

Table 1 on page 9:

Table 1. Statistics of the Panel

Scenario

Statistics

Mean

Median

Maximum

Minimum

Std. Dev.

Jarque-Bera

Observations

3h

TSC

3118.17

2142.47

18044.96

0.00

3232.37

6236.25***

22200

SE

1264.25

800.20

13293.21

0.00

1625.02

34512.37***

22200

TC

1853.92

1201.03

9603.15

0.00

2080.01

6675.69***

22200

TD

6.29

3.78

31.43

0.00

6.47

6391.08***

22200

4h

TSC

5243.53

3759.08

32524.24

0.00

4950.03

7569.96***

23400

SE

2877.62

1980.90

24859.88

0.00

3079.37

21864.19***

23400

TC

2365.91

1695.00

13091.84

0.00

2508.46

4987.88***

23400

TD

6.23

3.73

31.43

0.00

6.42

6994.53***

23400

5h

TSC

7834.25

5907.43

39346.86

0.00

7153.65

6117.85***

23736

SE

2643.96

1515.24

28129.41

0.00

3290.99

52697.42***

23736

TC

5190.29

3874.17

27821.81

0.00

4684.40

5322.63***

23736

TD

6.23

3.73

31.43

0.00

6.42

7106.32***

23736

Notes: ***Indicates significance at the 1% level, and a balanced dataset during the period of 2008-2019.

Table 2-6 on page 13-14:

Table 2. Results of panel unit root test for different scenarios

Variable

ADF-Fisher Chi-square

Level

First difference

Intercept

Intercept and trend

Intercept

Intercept and trend

3 Hours scenario

lnDT

9705.94***

9938.05***

24482.8***

18479.2***

lnSTC

5471.70***

7883.46***

16976.4***

12788.4***

lnSE

5111.76***

5962.00***

12993.2***

9253.21***

lnTC

5140.51***

6652.29***

14961.4***

10856.1***

4 Hours scenario

lnDT

10277.2***

10521.3***

25826.6***

19533.2***

lnSTC

9051.59***

11758.8***

20358.8***

15856.7***

lnSE

8656.93***

10279.0***

18331.8***

13020.2***

lnTC

6856.21***

8538.02***

16729.3***

12625.7***

5 Hours scenario

lnDT

8334.73***

7766.85***

21767.9***

16248.4***

lnSTC

13759.1***

11003.3***

18474.4***

15547.3***

lnSE

9107.27***

8840.42***

15808.2***

11801.4***

lnTC

9360.46***

7714.51***

15238.3***

12513.8***

The optimal lags are selected automatically by Schwarz information criteria. Notes: ***p< 0.01, **p<0.05, *p< 0.1

Table 3. Results of Pedroni cointegration test

Scenario

3 Hours scenario

4 Hours scenario

5 Hours scenario

Alternative hypothesis: common AR coefs. (within-dimension)

Panel v-Statistic

-6.54

-8.43*

-15.43*

Panel rho-Statistic

10.98**

21.29

29.33

Panel PP-Statistic

-62.70***

-28.95***

-17.55***

Panel ADF-Statistic

33.95***

36.10***

16.88***

Alternative hypothesis: individual AR coefs. (between-dimension)

Group rho-Statistic

31.83

36.76

39.60

Group PP-Statistic

-64.31***

-74.07***

-75.12***

Group ADF-Statistic

30.68***

29.57***

19.43***

Notes: ***p< 0.01, **p<0.05, *p< 0.1

Table 4. TSC effect of HSR on destination development (3h)

Explanatory

(1)

(2)

(3)

(4)

variables

lnDT

lnDT

lnDT

lnDT

lnTSC

-0.193***

(0.0104)

lnSE

-0.197***

-0.0763***

(0.0141)

(0.0209)

lnTC

-0.196***

-0.170***

(0.0119)

(0.0168)

Cons

3.076***

2.950***

3.005***

3.339***

(0.0828)

(0.103)

(0.0901)

(0.126)

Year FE

YES

YES

YES

YES

TDR FE

YES

YES

YES

YES

Observations

14594

11665

12874

9945

R-squared

0.026

0.019

0.023

0.029

F-test

341.4***

194.4***

271.4***

127.5***

Notes: Dependent variable: lnDT. Standard errors in parentheses, * p < 0.1, ** p < 0.05, and *** p < 0.01. The estimation approach selection is based on the Hausman test.

Table 5. TSC effect of HSR on destination development (4h)

Explanatory

(1)

(2)

(3)

(4)

variables

lnDT

lnDT

lnDT

lnDT

lnTSC

-0.117***

(0.00767)

lnSE

-0.116***

-0.0449***

(0.00851)

(0.0144)

lnTC

-0.175***

-0.152***

(0.00992)

(0.0140)

Cons

2.547***

2.472***

2.921***

3.093***

(0.0642)

(0.0665)

(0.0774)

(0.0913)

Year FE

YES

YES

YES

YES

TDR FE

YES

YES

YES

YES

Observations

16714

15535

13954

12775

R-squared

0.015

0.013

0.025

0.027

F-test

232.2***

185.6***

311.3***

154.5***

Notes: Dependent variable: lnDT. Standard errors in parentheses, * p < 0.1, ** p < 0.05, and *** p < 0.01. The estimation approach selection is based on the Hausman test

Table 6. TSC effect of HSR on destination development (5h)

Explanatory

(1)

(2)

(3)

(4)

variables

lnDT

lnDT

lnDT

lnDT

lnTSC

-0.0910***

(0.00651)

lnSE

-0.110***

-0.0958***

(0.00828)

(0.0143)

lnTC

-0.124***

-0.0588***

(0.00771)

(0.0149)

Cons

2.358***

2.412***

2.607***

2.801***

(0.0566)

(0.0650)

(0.0645)

(0.0850)

Year FE

YES

YES

YES

YES

TDR FE

YES

YES

YES

YES

Observations

17815

13952

17171

13308

R-squared

0.012

0.014

0.017

0.019

F-test

195.3***

178.1***

260.6***

116.9***

Notes: Dependent variable: lnDT. Standard errors in parentheses, * p < 0.1, ** p < 0.05, and *** p < 0.01. The estimation approach selection is based on the Hausman test.

Interestingly, the effect coefficients of TSC effect of HSR on destination development manifest a descending pattern, from -0.193 in scenario 1 (3 hours), to -0.117 in scenario 2 (4 hours), and then to -0.091 in scenario 3 (5 hours). The effects of TC and SE by HSR follow similar descending patterns. For instance, TC effect of HSR negatively affect destination development, from a coefficient of -0.196 in scenario 1 (3 hours), to -0.175 in scenario 2 (4 hours), and then to -0.124 in scenario 3 (5 hours). The impact coefficients of TC effect of HSR on destination development within temporal distance (3 hours, 4 hours and 5 hours) are -0.197, -0.116 and -0.110 respectively. In another word, the farther the temporal distance is, the weaker the inhibitory effect is. Considering both TSC and SE effect of HSR, their inhibitory effects significantly reduce when the temporal distance increases from 3 hours to 4 hours.

Further, TSC and SE effect of HSR on destination development show similar impacts across three temporal scenarios. However, for TC and SE effect of HSR, (1) TC effect of HSR (β = -0.170, p. < 0.01) illustrates a relatively stronger inhibitory effect than SE effect of HSR (β = -0.076, p. < 0.01) in scenario 1 (3 hours); (2) TC effect of HSR (β = -0.152, p. < 0.01) illustrates a significantly stronger inhibitory effect than SE effect of HSR (β = -0.045, p. < 0.01) in scenario 2 (4 hours); and (3) TC effect of HSR (β = -0.059, p. < 0.01) illustrates a relatively weaker inhibitory effect than SE effect of HSR (β = -0.096, p. < 0.01) in scenario 1 (5 hours).

Reviewer 2 Report

N/A

Author Response

There are no new suggestions for us, thank reviewer No.2 for his/her affirmation to us.

Round 3

Reviewer 1 Report

I agree to publish this paper